# Numerical Study of an Oscillating-Wing Wingmill for Ocean Current Energy Harvesting: Fluid-Solid-Body Interaction with Feedback Control

**David Balam-Tamayo [1], Carlos Málaga [2]** and **Bernardo Figueroa-Espinoza [1,*]**

[1] Instituto de Ingeniería, LIPC, Universidad Nacional Autónoma de México, Sisal, Yucatan 97355, Mexico; dbalam@hotmail.com

[2] Facultad de Ciencias, Universidad Nacional Autónoma de México, Mexico City 04510, Mexico; cmi.ciencias@ciencias.unam.mx

[*] Correspondence: bfigueroae@iingen.unam.mx

**Abstract:** The performance and flow around an oscillating foil device for current energy extraction (a wingmill) was studied through numerical simulations. OpenFOAM was used in order to study the two-dimensional (2D) flow around a wingmill. A closed loop control law was coded in order to follow a reference angle of attack. The objective of this control law is to modify the angle of attack in order to enhance the lift force (and increase power extraction). Dimensional analysis suggests a compromise between the generator (or damper) stiffness and actuator/control gains, so a parametric study was carried out while using a new dimensionless number, called *B*, which represents this compromise. It was found that there is a maximum on the efficiency curve in terms of the aforementioned dimensionless parameter. The lessons that are learned from this fluid-structure and feedback coupling are discussed; this interaction, combined with the feedback dynamics, may trigger dynamic stall, thus decreasing the performance. Moreover, if the control strategy is not carefully selected, then the energy spent on the actuator may affect efficiency considerably. This type of simulation could allow for the system identification, control synthesis, and optimization of energy harvesting devices in future studies.

**Keywords:** oscillating foil; closed loop controlled oscillating foil; fluid-structure interaction; wingmill

## 1. Introduction

Energy harvesting from water currents in ocean, rivers, or channels has been extensively investigated in the past [1,2]. However, it is still a very active field of research due to the large energy density of sea currents (1000 times more than air per meter squared, for the same current speed). The most conventional among the many schemes that have been proposed is the use of horizontal axis turbines [3,4], similar to the well known horizontal axis wind turbines (HAWT) that were used in wind farms on both land and offshore sites. Even though the oceanic conditions make the commercial use of current energy very difficult due to the harsh sea/rivers environment, the first principles are the same (as in HAWT), and are well known. The idea of harvesting energy with oscillating foils (called wingmills) was first presented by Mckiney and D'Lauriel in 1981, [5]. The principle of operation is simpler than that of turbines. The oscillating foil presents some advantages with respect to conventional rotating configurations: from the point of view of its ecological impact, the motion of the foil is more uniform and slower, so marine life would have less probability of harm when swimming close to the device [6,7]. Moreover, some studies suggest that the efficiency that can be reached with oscillating foils may be comparable to those of wind turbines [7–10]. Currently, many simulations and prototypes are being developed motivated by the high efficiency possibilities and other potential advantages. An example is the Stingray Tidal Stream, which is a full scale prototype that is used to

harvest tidal current energy. The prototype was tested in the UK in 2002 [11–13]. To our best knowledge, there are only three executive summaries that are available to the public. Apparently there was an economic feasibility issue, which was probably due to low overall efficiency (of the order of 11.5%).

From the research point of view, there have been many significant advances and estimations of the efficiency for the case of oscillating foils, both numerical and experimental. Comprehensive reviews can be found in [14,15]. Experimental investigations have also been carried out by [8,9,16,17] while using different layouts; an interesting example can be found in Kinsey et al. [8], who tested a tandem passive configuration, where the foil trajectory was given by a sensible design of a four-bar mechanism. The results showed the measured efficiencies that can reach as much as 40%. With this configuration, the pitching and heaving motion of the foil are restraint by a four-bar mechanism. Some experimental and numerical simulation investigations are summarized in Tables A1 and A2, in Appendix A, based on the work of Young et al. (2014) [14].

Oscillating foils can be classified according to the use of additional energy input other than the fluid flow: passive devices would correspond exactly with the self-sustained devices previously mentioned [14,15]. Conversely, active devices can be implemented as: (a) closed loop control, where there is a pre-defined control objective (maximum lift, stability, efficiency, etc). In run-time, a relevant output is measured and then fed-back to a decision maker (the controller), who assigns a control action based on the error. This error-dependent response is sent to an actuator (a motor applying torque, for example), which drives the dynamics of the system in order to comply with the control objective. (b) Open loop: there is energy input, but no feedback. For example, a sinusoidal torque is applied to the pitching motion. The control action does not depend on an error measurement.

The reader may not confuse the term "Active Control", which is widely used in the literature with "Feedback Control". Other research groups [18–21] have implemented simulations of actively controlled foils; however, their control schemes are actually open-loop.

There is a knowledge gap in the literature concerning the effect of feedback control on oscillating foils; most investigations that contain all of these elements are related to aeroelasticity and wind energy in HAWTs. One particular case where a similar scheme was implemented is [22], where a PID controller was coupled to a CFD model for flutter control purposes. We also found a patent in the literature [23], where this scheme is considered for commercial use. However, this patent is very general and it does not contain any engineering details of the implementation. The high level of complexity of the fluid-solid-body interaction hinders the development of effective closed-loop control strategies for these configurations. Moreover, there is still no analytical model that allows for the synthesis of a control law that is optimal in terms of energy harvesting.

The present investigation proposes the implementation of a fully coupled CFD-solid-body interaction while using a closed loop feedback control scheme on an oscillating foil. A very simple control law was implemented that aims to set the angle of attack in such a way as to have maximum lift. The actuator applies torque at the center of rotation of the foil, which is proportional to the difference between a desired and the measured angle of attack. Note that this choice of angle of attack cannot (yet) be proved to be optimal in terms of efficiency. This work will focus on the effects of implementing a feedback controller in the complex fluid-structure interaction framework of a wingmill.

## 2. Problem Statement

The power extraction system is composed by an airfoil (NACA0015) of chord length $c$, whose solid body motion is constrained to pitching and heaving. The center of mass of the foil moves along the $y$ axis, as shown in Figure 1. The vertical motion is constrained to the range $-h_0 < y < h_0$, so the airfoil should cycle between those limits in order to work continuously. The center of rotation, as well as the center of mass, are located at the chordline, at a distance $0.4c$ from the leading edge. The pitching angle $\theta$ can vary in the

range $0 \leq \theta < 2\pi$. The airfoil is immersed in a fluid of constant density $\rho$ and viscosity $\mu$. The velocity field far from the airfoil is uniform, with speed $U$ along the $x$ axis (horizontal). An actuator (representing a motor) can apply a torque $T$ at the center of rotation, which causes the airfoil to pitch. The airfoil is neutrally buoyant in this case (the scheme can also handle buoyancy, but we omitted buoyancy for simplicity).

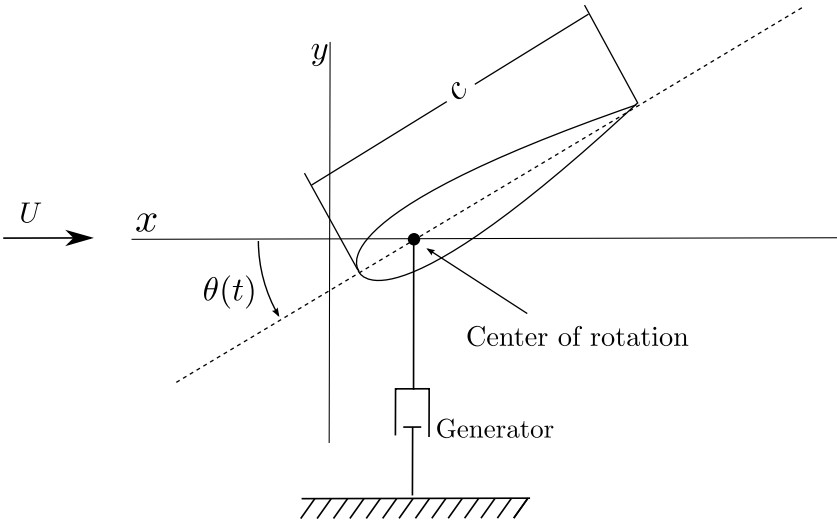

**Figure 1.** Scheme of the active model implementation.

The equations of motion for the airfoil are Newton's second law applied to the forces in the $y$ direction, as well as the torques in the $\theta$:

$$M\ddot{y} = -b\dot{y} + \mathbf{e_y} \cdot \int_S \sigma \cdot \mathbf{n}\, dS \tag{1}$$

$$J\ddot{\theta} = -b_\theta\dot{\theta} + \mathbf{e_y} \cdot \int_S \mathbf{r}_S \times (\sigma \cdot \mathbf{n})\, dS + T \tag{2}$$

$$T = Ke \tag{3}$$

$$e = \alpha_{ref} - \theta \tag{4}$$

where $M$ and $J$ are the airfoil's mass and moment of inertia, $\mathbf{e_y}$ is the unit vector in the $y$ direction, $\sigma$ is the fluid stress tensor, $\mathbf{n}$ is the unit normal to the surface, and $\mathbf{r}_S$ is the vector that goes from the center of rotation to a given point on the foil surface. The surface integrals shown in Equations (1) and (2) represent the hydrodynamic and viscous forces that couple the solid body and fluid motion. The coefficient $b$ is an equivalent damping constant that represents the vertical force due to a linear generator extracting energy from the device, proportional to the vertical component of the velocity $\dot{y}$. Finally, $b_\theta$ is a damping coefficient for the foil pitching. In our particular case, this parameter is very small, and its contribution to the motion is negligible.

*The Feedback Loop*

The torque $T$ that is applied at the center of rotation is the control action, and it is proportional to the error $e$ with respect a reference angle $\theta_{ref}$. An experimental prototype would use an actuator, i.e., a DC motor or a servo. The gain $K$ that is applied to the error represents the effect of such an actuator combined with the proportional control law (when considering the actuator torque to be proportional to the electrical current supplied by the controller, as is the case for many servo-motors).

The choice of the reference angle $\theta_{ref}$ is not a trivial task. For the moment, the optimal control problem is out of the scope of this work, given the absence of a simple model that would allow for a controller synthesis. Instead, we decided to propose a heuristic approach:

we propose setting $\theta_{ref}$ in such a way as to maximize the Lift Coefficient $C_L$ of the airfoil (per unit span, since the flow is 2D), which is defined as:

$$C_L = \frac{F_L}{\frac{1}{2}\rho U^2 c} \tag{5}$$

where $F_L$ is the Lift Force, which depends on the angle of attack $\theta$ and the Reynolds number, defined as:

$$Re = \frac{\rho U c}{\mu} \tag{6}$$

There is a maximum $C_L$ that can be estimated from the well known aerodynamic theory for airfoils (polar diagram) [24,25]. These polar diagrams contain the maximum lift that can be attained (before the airfoil stalls), which corresponds to a "static" angle of attack $\theta_{ref}$ (for a given $Re$). Let us note that these diagrams were obtained under steady conditions, which is not the case in an oscillating foil. However, we had to use this maximum lift (and the corresponding $\theta_{ref}$) as a first approximation of the maximum unsteady lift conditions. In this particular case, the reference static angle of attack (that corresponds to our particular Reynolds number) is approximately 0.310 rad  (17.76 deg). We chose a value 5% smaller in order to avoid stall, giving $\theta_{ref} = 0.295$ rad (16.90 deg).

We call this reference static, because, when the airfoil is heaving, the actual angle of attack $\alpha$ must be estimated in terms of the relative velocity of the foil with respect to the fluid, which results in a correction that can be expressed as [17,26]:

$$\alpha(t) = -\arctan\left(\frac{\dot{y}}{U}\right) + \theta(t) \tag{7}$$

in the moving reference frame, where $\theta(t)$ is the (measured) angle between the airfoil chord and the $x$ axis. Both $\alpha(t)$ and $\theta(t)$ are time dependent. Following this scheme, the error $e(t)$ is estimated from the comparison of the reference and measured (and corrected) angles of attack:

$$e(t) = \theta_{ref} - \alpha(t) = \theta_{ref} + \arctan\left(\frac{\dot{y}}{U}\right) - \theta(t) \tag{8}$$

In terms of the measured angle $\theta(t)$, one can define a dynamic reference

$$\alpha_{ref}(t) = \theta_{ref} + \arctan\left(\frac{|\dot{y}|}{U}\right)\text{sign}(\theta_{ref}) \tag{9}$$

such that $e(t) = \alpha_{ref} - \theta(t)$. This error (based on a fixed reference frame) was used by the Proportional Controller in the Feedback Loop (and it will be used in what follows to assess the control scheme). Note that the argument of the arctan function in the correction for the moving reference frame is always positive. When the reference angle is negative, one has to correct with a negative value (note also that $\dot{y}$ may have a different sign than $\theta_{ref}$). Due to the finite height of the channel, the reference angle of attack has to change direction; this happens when the heaving position reaches the switching point $y = \pm h'$, where $h'$ was chosen accordingly, so that the airfoil can switch direction without overshooting the prescribed heaving limit $h_0$ (reaching $y = \pm h_0$ with zero vertical speed $\dot{y} = 0$).

### 3. Numerical Simulations

OpenFOAM is the code used to solve the problem. It is an open source code that uses the Finite Volume Method to solve the Navier–Stokes equations of fluid motion, as well as other partial differential equations of mathematical physics. More information about the code is readily available at [27–30] The numerical method used for the windmill case [31] is coded in a solver, called pimpleDymFoam (https://openfoamwiki.net/index.php/PimpleDyMFoam), and it is second order accurate in time and space. The case required the use of moving mesh capability [32–34], as well as solid-body interaction with

the fluid [35]. The method for handling the moving mesh is explained in detail in [36]; we selected a mesh deformation method that uses the Laplace equation (with variable diffusion) in order to determine the motion of each mesh point, while preserving the motion that is given by the moving object. This methodology can cope with large deformations while preserving the mesh quality. However, errors may be introduced due to highly distorted cells, particularly when angular deformations are present. This method is well suited for efficient coupling with Finite Volume Method (FVM) solvers [31,36].

The fluid-body coupling is done through an OpenFOAM module, called SixDOF (six degrees of freedom rigid body motion). The details of this module can be found in [37]. Modifications to this part of the code allowed for the introduction of the feedback loop. The SixDOF VOF-solver uses a series of coordinate transformations and axis translations to the center of mass and rotation of the moving object. However, the most delicate part of the coupling lies on the PIMPLE algorithm (Semi-Implicit Method for Pressure Linked Equations) [31]; the rigid body motion is solved by a routine that is similar to the leapfrog method; however, an improved explicit-implicit scheme that is based on the second order Adams–Bashforth–Moulten formulation provides an outer loop of predictor-corrector steps to achieve convergence (based on the motion of two previous steps). Aitken's under-relaxation constants are adjusted dynamically based on the acceleration of the rigid body, thus improving stability. This type of simulations with deformable grids, applied to fluid-structure interaction using OpenFOAM solvers have been tested in the past with complex problems, such as drag reduction [38], flow and deformation in arteries [39], bluff body oscillations [40], or offshore floating platforms [41]).

The code was modified in order to compose a case where the airfoil, the fluid, and a feedback control loop are coupled through the equations of motion (1)–(4). These modifications allowed for the study of the effects that the feedback control law has on the windmill energy harvesting (from a fluid current).

*3.1. The Mesh*

The mesh was created from an STL file while using the definitions of NACA four digits profiles [42] (there exist many online tools to create NACA airfoils; we used a CAD software in order to generate the stl). The external mesh was composed while using OpenFOAM tools that help define the different parts of the mesh and its relationships (blockMesh), as well as selectively refining, using a particular set of tolerances and user-adjustable mesh parameters (SnappyHexMesh) [27]. The resulting mesh is composed of several regions with different levels of refinement. The highest level of refinement is a small region that surrounds the surface of the airfoil, as shown in Figure 2. It is contained in a larger region (deformable mesh), which can deform continuously while complying with the solid body motion of the internal mesh. This deformable mesh is circular (in this case) and is itself contained in a rectangular region, called refinement box, lying inside the principal mesh, which follows with decreasing refinement until the domain boundaries are reached. The main disadvantage of this scheme is that large deformations of the mesh can cause numerical problems (largely deformed cells). This limits the final displacement of the airfoil. Additionally, the rotation of the body is limited to approximately 90 degrees (in each sense of rotation). In spite of these limitations, this setup allowed for the simulation of the wingmill and the control scheme. Figure 2 shows the aforementioned regions and the dimensions of the mesh. Each mesh refinement step divides the mesh cell width and height by two. All of the simulations had a minimum mesh width, such as to fit at least four mesh cells inside the boundary layer near the airfoil surface.

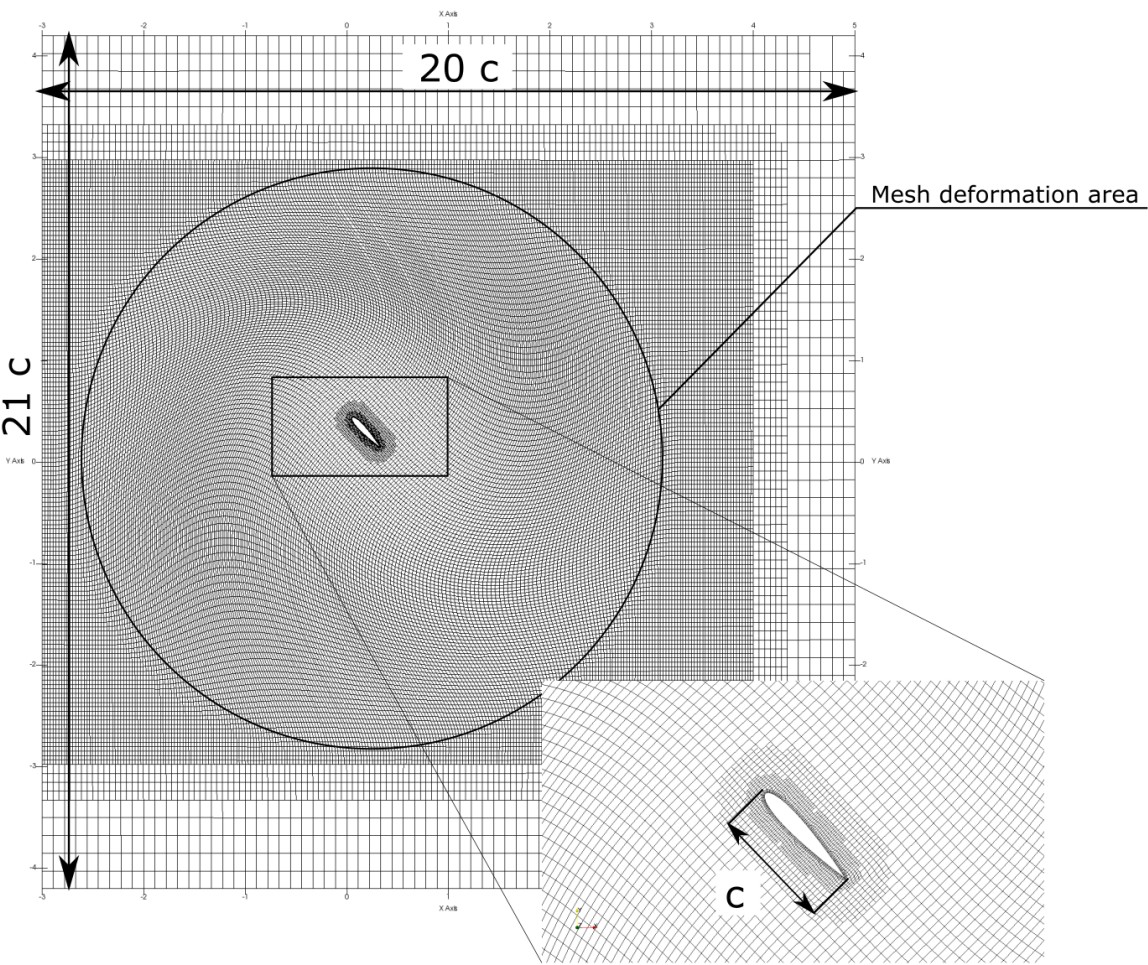

**Figure 2.** Mesh configuration: to the center, the airfoil and the first two internal mesh regions, which are are zoomed-out in the box to the lower right. The circular mesh deformation area is also shown (deformable mesh). The domain boundaries are indicated in units of chord size. The deformable region is contained in a refinement region, which can contain different levels of refinement.

*3.2. Mesh Validation*

The resulting movement of the foil describes a quasi-periodic trajectory in the sense that the cycles are not all equal. This causes that the outputs (energy extraction in our case) must be averaged in order to test convergence. The first (two or three) cycles showed the effect of transients, which tend to disappear as time passes, so later cycles tend to converge in time. This average power $\bar{P}$ is obtained by integrating the net energy (extracted minus control energy) and dividing by the corresponding time that elapsed. This gives the average energy that we used for the calculation of efficiency. Figure 3 shows the evolution of this validation parameter as a function of the number of cycles. The simulation time was selected, such as to guarantee that the variation of the average output is below 0.5 percent. This simulation time resulted in being of at least 25 cycles. The efficiency of power generation is then calculated as the ratio between the average power output and the total Power of the flow (through the frontal area that is swept by the device):

$$\eta = \frac{\bar{P}}{\frac{1}{2}\rho U^3 s h_0} \tag{10}$$

where $s$ is the span of the foil (and $sh_0$ the swept area). In 2D airfoils, the average power is per unit span.

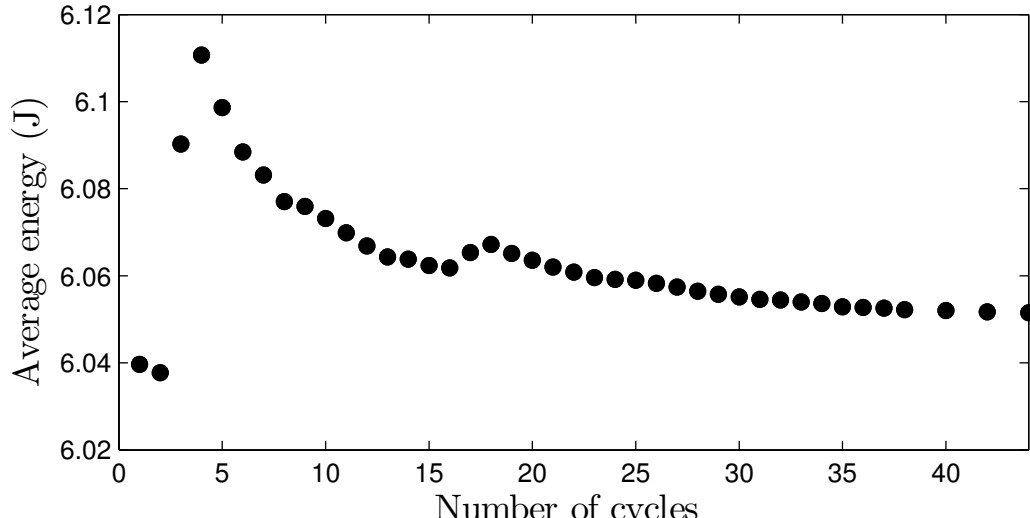

**Figure 3.** Variability of average energy production as a function of the number of cycles taken into account for averaging.

The complex structure of the mesh makes its validation rather cumbersome. There is a compromise between the refinement level of the mesh and the extent of the mesh deformation. Highly refined meshes are more susceptible to large angular deformation of the cells (squares deform into rhomboids with very large aspect ratio), which causes numerical errors. Thus, the level of refinement is limited. Six configurations with different levels of refinement were tested. Five of them ended the tests without crashes or errors. The variation of the averaged energy output is smaller than 1 percent (see Table A3).

No-slip boundary conditions were imposed on the fluid velocity at the foil surface, while, on the top and bottom of the mesh, a symmetry (no penetration) condition was set. Far upstream the leading edge an input boundary condition was imposed, with a uniform velocity profile with speed (in the $x$ direction) $U_\infty$. To the right, far from the trailing edge, an exit boundary condition was implemented. The distance from the foil to the domain boundary was also validated. As the domain grows larger, the validation parameter converged to within 0.5% of the final value, for a domain of $21c$ width and $20c$ height. The final selection was M5 from Table A3, which showed a reasonable compromise between the resolution and computational effort.

### 3.3. Control Scheme

The control strategy is based on the reference angle of attack $\theta_{ref}$, which comes from the $C_L$ vs. $\theta$ plots, called polars (there is one curve for each Reynolds number). These polars come from experiments on steady flow [43]: typically, the lift increases monotonically with $\theta$ for small angle of attack, until a maximum lift is reached. Larger angles of attack (AoA) would cause stall (flow separation and formation of recirculation eddies behind the detachment point), causing the lift to drop sharply. If we assume that this relationship between lift and angle of attack is the same for the unsteady case, a heuristic approach would intend to maintain the maximum lift angle of attack $\theta_{ref}$ in order to achieve maximum power extraction. Because the airfoil is moving, this AoA should be corrected due to the heaving motion with Equation (9), so the controller has to do calculations "on the loop" in order to know this dynamic angle of attack $\alpha_{ref}$. In reality, the unsteady lift results in being very different to the one described above; it presents non-linearities and hysteresis loops that become very complicated with the online corrections for the dynamic angle of attack, not to mention the effects of the feedback control [44,45].

Because $\pm h_0$ limits the heaving motion, the motion must be periodic. An optimal controller should reach $\alpha_{ref}$ as fast as possible and stay there until the foil has to change direction (switching sign at $y = h'$, reaching $h_0$ some instants later). Clearly, this heuristic control strategy is not optimal, since there is an additional complication: the cost of controlling the AoA is important. The optimal controller must maximize power ($C_L \dot{y}$) and, at the same time, minimize the control energy $T\dot{\theta} = (Ke)\dot{\theta}$, as in Equation (3). An interesting optimal control problem that is out of the scope of the manuscript, but it can be addressed with the help of coupled closed-loop simulations, like the ones proposed here.

Figure 4 shows the control scheme. It is composed by four principal blocks: 1. the Plant, which comprises the CFD solver and an OpenFOAM module SixDOF [37]. This module couples the CFD simulation with the rigid body motion Equations (1) to (4) (in our case, we have only two degrees of freedom); 2. The Control Loop was implemented by modifying the code in order to add an external angular torque $T$ to the rigid body motion according to Equation (3); 3. An additional Data Extraction module was coded, which is used as a virtual representation of a set of sensors, and 4. The Reference Angle computation module calculates the ($\alpha_{ref}$) from Equation (9). All of these interactions occur within the OpenFOAM code, so the solid body and the fluid dynamics (as well as the control action) are coupled.

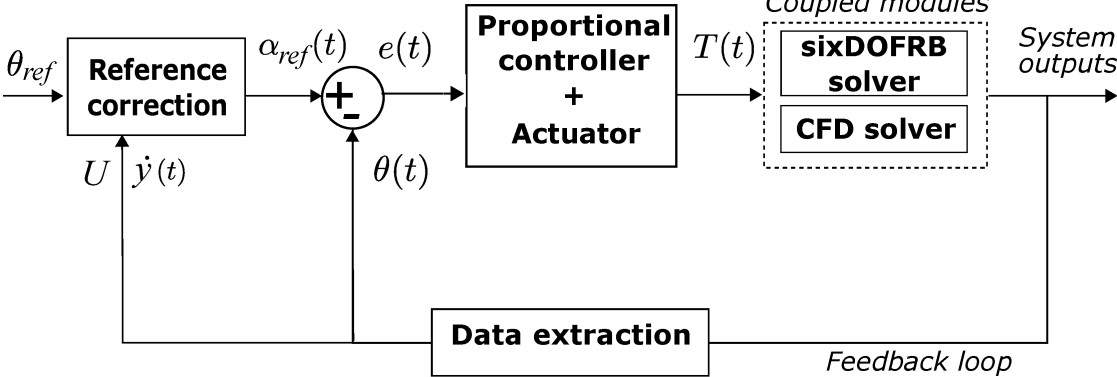

**Figure 4.** Close loop active control scheme. Dashed box groups the coupled modules that conform the equivalent to the plant of the system. $T(t)$ stands for torque.

### 3.4. Limitations

There are additional restrictions that must be observed while using this fully coupled feedback control scheme: As already mentioned in previous sections, the deformable mesh limits both the heaving and pitching motion of the foil. The mesh is very dense near the airfoil surface and the cell size increases stepwise in regions far apart from the foil. The smaller cell size near the foil, in general, led to more refining steps. The best combinations resulted in a similar number of cells in spite of having very different refinement levels at the first cell (Table A3). We also noted that a dense mesh far from the foil implies very deformed cells (due to mesh motion), which may bring numerical problems (as well as large processing time). That is why there is a compromise between those parameters in order to obtain a consistent mesh to test all of our cases. Secondly, the foil is restricted to travel vertically a distance $h_0$ on each direction. The feedback loop that we implemented is single-variable, i.e., it controls the pitch to maximize the lift, but not the displacement in $y$. The maximum displacement $h_0$ was adjusted by using a displacement $h'$ that has to be reached (before reaching $h_0$), causing the sign of the reference angle of attack to change. Note that the simulation scheme allows for the implementation of a multi-variable (MIMO) control. The proposed control law has not yet probed to be optimal, as already mentioned.

## 4. Dimensionless Parameters

The problem of the oscillating foil, although apparently simple, represents a very complex problem: there are many variables involved. For instance, the liquid properties $\rho$ and $\mu$, the airfoil density $\rho_a$, the current speed $U$, the generator equivalent damping constant $b$, two characteristic lengths: foil chord $c$ and the breadth of the heaving motion $h_0$, the controller-actuator gain $K$, translational and angular inertia: $m$ and $J$, the heaving frequency of the resulting motion $f$, and the average power generated $\bar{P}$. The Buckingham $\Pi$ theorem gives nine dimensionless numbers:

1. The efficiency $\eta = \frac{\bar{P}/s}{1/2\rho U^3 h_0}$
2. The Reynolds number $Re = \frac{\rho U c}{\mu}$
3. Dimensionless heaving $\frac{h_0}{c}$
4. Inertia 1 $\Pi_1 = \frac{M}{\rho c^3}$
5. Inertia 2 (rotational) $\Pi_2 = \frac{J}{\rho c^5}$
6. Dimensionless damping ratio $\Pi_3 = \frac{b}{\rho c^2 U}$
7. Dimensionless gain $\Pi_4 = \frac{K}{\rho U^2 c^3}$
8. Strouhal number $St = \frac{fc}{U}$
9. Density ratio between the fluid and solid body $\frac{\rho}{\rho_a}$

where $\bar{P}/s$ in $\eta$ stands for power per unit wingspan. In this investigation, $Re$, $\Pi_1$, and $\Pi_2$ are constants that are given by the problem statement. The other parameters $\Pi_3$, $\Pi_4$ (generator "stiffness" and controller-actuator gain) can be set before the simulation. The output of such simulation would be a flapping frequency $St$, a heaving breadth $\frac{h_0}{c}$, and efficiency $\eta$. In our particular case, we were able to tune-up the controller in order to obtain the desired values of $\frac{h_0}{c}$ in any case (within the limits of the simulations given in Section 3.4). Note that the density ratio between liquid and airfoil density $\rho/\rho_a$ is an important dimensionless number that influences the dynamics of oscillating foils. In our case, this parameter equals one. Our simulations can handle differences in density; however, for this particular manuscript, we did not want to add more complexity to the discussions.

Preliminary results showed that when the generator is very "stiff" in the sense that the force of the generator ($b\dot{y}$) is large with respect to the lift (large $\Pi_3/C_L$, where $C_L$ is the Lift Coefficient), the foil moves very slowly and the efficiency drops. On the contrary, if the generator is too soft, then the force will be too small to produce power and the efficiency will be low as well. Conversely, if the controller-actuator gain $K$ is large, the torque will also be large and the response of the system in terms of $\theta$ will be fast, but the energy expense will also be large, affecting the efficiency (large $\Pi_4/C_L$). If $\Pi_4$ is too small, then $\theta$ will be too slow to follow the reference, $St$ will be small, and the efficiency will also drop. One can search for a new parameter that combines these two effects, comparing the torque that is caused by the controller-actuator and the stiffness of the generator, by simply dividing

$$B = \frac{\Pi_3}{\Pi_4} = \frac{bUc}{K} \tag{11}$$

Numerical experiments show a maximum on the curve $\eta$ vs. $B$, for different values of $h_0/c$. These results are presented and discussed in the following sections.

### Reynolds Number and Turbulence

All of the cases simulated in this work are dominated by inertia; however, the reader must note that the boundary layer transition to turbulence for flows around thin airfoils occurs at Reynolds numbers in the range $5 \times 10^4 < Re < 2 \times 10^6$ [46–48]. Some phenomena, like dynamic stall, may occur well below the transition to turbulence. The dimensional numbers matrix for the simulations can be consulted in Section 3.4 and Table A4.

Three different types of test cases were considered for this study: (i) Low Reynolds number: $Re = 1 \times 10^3$. This may result in unrealistic cases, but the purpose was simply to test the consistency of the coupled system behavior, and the effect of the control loop on the system dynamics, without turbulent effects. (ii) High Reynolds number, no turbulence model: $Re = 2 \times 10^4$. This type of simulations shows the performance of the control scheme with strong inertial effects. The scaling of important variables, such as efficiency, will be discussed, as well as important phenomena, such as boundary layer separation, eddy detachment, or stall. (iii) High Reynolds number, identical conditions, as above, but with turbulence model: $Re = 2 \times 10^4$. A RANS-based turbulence model (k-omega-SST) was chosen, which can be used as a Low-Re turbulence model, without requiring extra damping functions. Far from the boundaries, it behaves as a $k - \epsilon$ (free-stream). Consequently, the model is relatively insensitive to free-stream turbulence properties. In general, it is known to behave correctly in adverse pressure gradients and separating flow [49]. The turbulent intensity at the inlet was low (TI = 0.5%). We had to use wall functions, with the first cell (next to the foil's surface) at $y+ = 7$. However the use of wall functions in our case means that the $k - \omega$ part is not being used. The model can correctly estimate wall shear stress for large $y+$ [50], when boundary layer separation or stall are not present. On the contrary, if they occur, the details of the flow could be described (if $y+$ were sufficiently small), but this would require further investigation and experimental (or DNS) validation [51]. The detailed structure of the flow when stall is present was left out of the scope of this investigation. Nevertheless, we believe that, for the cases where stall occurred (most of the time it did not happen), one has to be cautious and interpret the simulation results as a reasonable picture of the system's dynamics and energy harvesting.

In general, the discussions will refer to the high $Re$ case without turbulence, unless stated otherwise. The qualitative behavior of the system at high $Re$ with turbulence resulted in being very similar to that without turbulence, as will be shown in later sections. However, turbulence is a very complex subject, which deserves a much more comprehensive study by itself. Because of the particular scope of this work, turbulence will only be discussed briefly in terms of the effect that it has on the system's performance, and it shall be discussed more in detail in future work.

Regarding the low $Re$ case, it is expected to have very low efficiencies due to the strong effect of viscosity (energy that is spent by the active control). However, we expect the system behavior to be qualitatively similar to the high $Re$ cases, since $Re = 10^3$ is still an inertial flow.

## 5. Results

Let us first discuss the way a typical simulation takes place: Figure 5 shows three panels: the first panel (a) shows the reference AoA $\alpha_{ref}$ (solid line) as well as the "measured" AoA $\theta$, as a function of time (in seconds). Note that the error is the difference between these two angles, proportional to the control action. Panel (b) shows the instantaneous power (at the generator, gross Power) as a thick gray line, the controller-actuator power consumption (dashed-dotted line), and the net power (continuous black line), which is the difference between the last two curves (generated power minus control effort). The third panel shows the airfoil heaving motion, and the airfoil icons show the corresponding angular position in time. In Figure 5, time starts at $t = 230$ s, once the transient effects have passed. The foil is moving downwards, producing power, and $\theta$ is very similar to $\alpha_{ref}$ (solid curve panel a), with a negligible expense of energy. Before reaching $t = 232$ s, the foil has reached $y = -h'$ and the controller sets a new negative $\alpha_{ref}$ for maximum (upwards) lift, forcing the foil to turn before reaching $h_0$. At $t = 232$ s, the controller then tries to reach this new reference (see panel (a)), consuming a large amount of power for some instants (second peak), as shown in panel (b). The controller causes $\theta$ to approach $\alpha_{ref}$ again in $t = 233$ s and, consequently, the energy that is consumed by the controller tends to zero again (the lift is supposed to be close to its maximum at this moment). The foil is now moving upwards (c). At this point, the foil is producing the near-maximum power until it

reaches the threshold $y = h'$, causing the controller to set a new $\alpha_{ref}$ positive ($t \approx 236$ s), and the cycle starts over again. Note that the heaving trajectory resembles a saw-tooth, while the pitching angle trajectory is actually a succession of squared pulses, similar to the curves described in [52], where the saw-tooth trajectory was imposed. Additionally, note that the curves in panel (a) are the dynamic angles of attack (corrected); this is why they do not appear squared (it is clear from this discussion that the reference angle of attack $\theta_{ref}$ is squared).

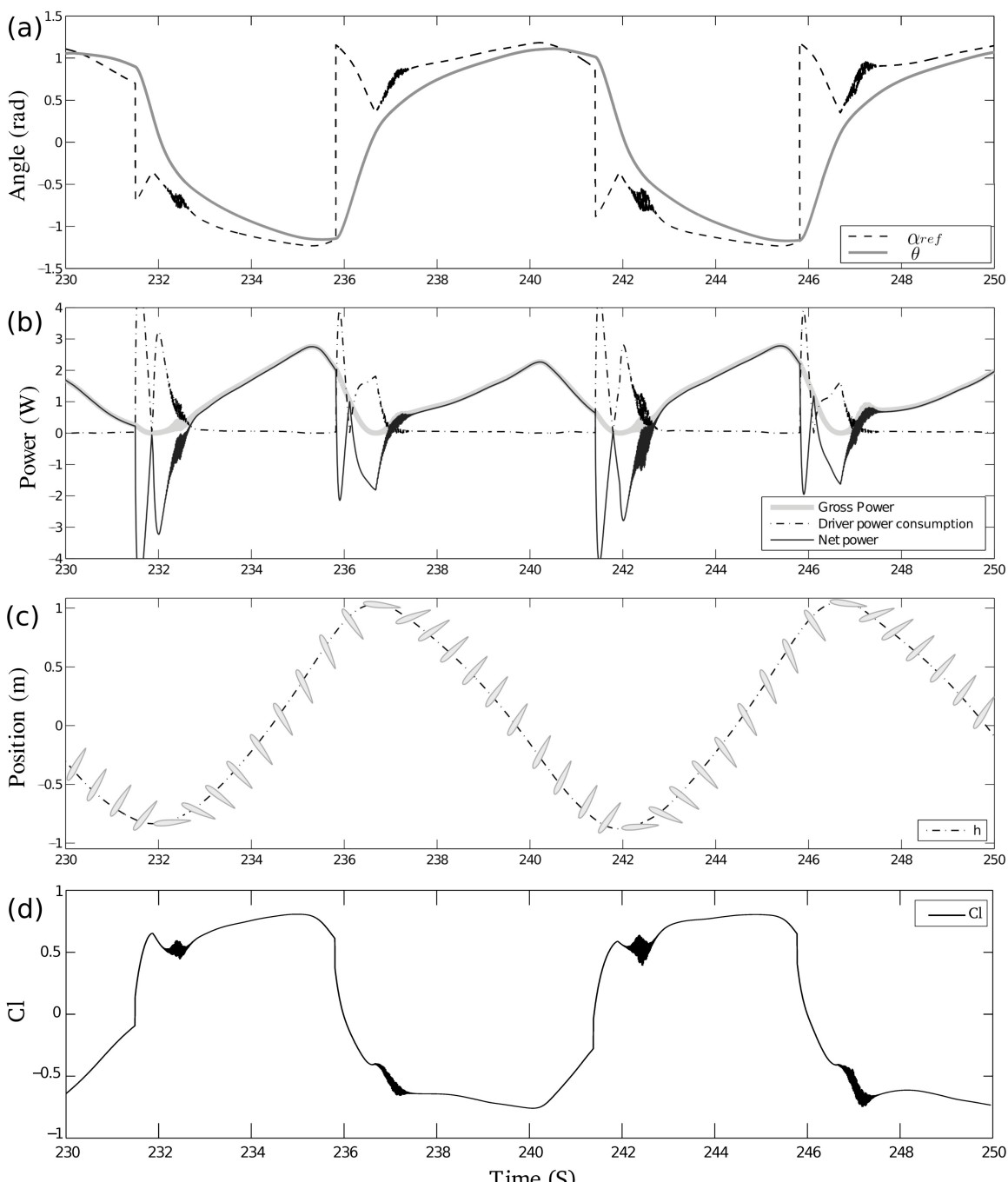

**Figure 5.** System response behavior. $h_0/c = 2.5$, $B = 0.375$. From top to bottom: (**a**) Reference tracking. (**b**) Power output and energy consumption. (**c**) Position of the rotation center of the foil. (**d**) Lift Coefficient $C_L$.

Figure A1 shows the same representation as in Figure 5, for $h_0/c = 1.25$ and $h_0/c = 3.25$. Note that the case $h_0/c = 1.25$ behaves similarly to $h_0/c = 2.5$, but the case $h_0/c = 3.25$

shows a very erratic power production. This is the result of the complexity that arises from the interaction between the solid body, the flow, and the closed loop control. The effect of the control scheme will be discussed next.

### 5.1. Effects of the Closed-Loop Control

There are some unexpected effects of the closed-loop scheme that became evident from these results. We enumerate some of them before commenting some very important effects on the efficiency:

1.  A noisy high frequency oscillation can occur on the reference signal for $\alpha_{ref}$ shortly after the foil has changed direction. This is an effect of the control action; since the dynamic angle $\alpha$ depends on $\dot{y}$, the latter derivative amplifies small oscillations that are caused by the control torque. Even if the torque is applied at the center of mass, linear and angular motions are coupled by hydrodynamic forces, so a sudden torque may cause a small jump in $\dot{y}$. This effect is only observable when the heaving speed is close to zero (and it can be avoided while using a properly tuned PI controller; this added complexity is out of the scope of the present report).

2.  The control effort (energy spent in the control action) may be very important in the vicinity of $y = h_0$. The net power could be substantially increased if this energy could be spared (following the sign change in the reference angle). Three possible ways of doing so would be: (a) a passive mechanism that turns the airfoil once the threshold $\pm h'$ is reached, (b) instead of pitching of the whole airfoil, set an aileron near the trailing edge, so the pitching effort decreases substantially, (c) to saturate the controller output to limit the spent energy (this would have consequences on the response time), and (d) to implement a control strategy that can be proved to be optimal in the sense of net power extraction. Of course, some of these suggestions may be combined in order to increase the efficiency.

3.  The power curve shows a maximum before reaching the time where the new $\alpha_{ref}$ is assigned. This should not happen if the lift coefficient was that of a static polar for the airfoil (maximum). The power should not decay until the controller sets a new reference ($h'$ is reached). Even if the reference angle was chosen, such as to have maximum static lift, the actual lift is not parallel to the $y$ axis, multiplying, by $cos(\Delta\theta)$, where $\Delta\theta = arctan(\dot{y}/U)$. This causes the lift to decrease whenever the heaving speed becomes comparable to $U$. When this happens, a plot of the factor $cos(\Delta\theta)$ versus power (not presented here for brevity) shows that the maximum power coincides with the cosine crest. This correlation is clear when $\dot{y}/U < 1$, as $cos(\Delta\theta) \simeq 1 - (\dot{y}/U)^2$ and the power that is extracted from the damping is proportional to $\dot{y}^2$. This is a very important consideration, because this is a limit to the velocity (and power) that can be extracted while using oscillating foils whose motion is constrained to the $y$ axis.

4.  Dynamic stall: even though the reference angle was chosen to be below the maximum lift (5% smaller) in order to avoid stall, a more detailed inspection of the boundary layer separation and vortices detachment made clear that unsteady stall is taking place for some cases. Figure 6 shows the vorticity of the velocity field for three cases ($B = 0.4$) at different times and $h_0/c$: (a) for $h_0/c = 2.5$, $t = 239$ s, the boundary layer remains attached to the foil, with the exception of a small perturbation (incipient eddy) forming near the leading edge lower surface. This eddy will eventually detach from the trailing edge. One can observe that a pair of counter-rotating vortices detached from the foil when it switched direction in the vicinity of $y = \pm h_0/c$; (b) for $h_0/c = 1.25$, $t = 239$ s the boundary layer clearly stays attached to the foil, and the only remnant eddies are again forming a vortex pair, close to the turning point $y = \pm h_0/c$; however, in (c), there is dynamic stall ($h_0/c = 3.75$, $B = 0.2$, $t = 235$). The image shows the airfoil heaving downwards (halfway towards $-h_0$). There is an alternating sign vortex street left from the previous cycle. There is clear detachment of the boundary layer in the lower surface of the airfoil. The time $t$ corresponds to the horizontal axis shown in Figure 5. The exact nature of the dynamic stall, in our

case, is out of the scope of this work; however, the reader may refer to [47,53,54] for a comprehensive characterization of the phenomenon. Moreover, we observed that, whenever the controller overshoots ($\alpha_{ref}$ crosses $\theta$), there is a detachment of eddies and oscillations in $\dot{y}$, so dynamic stall may, in some cases, be caused by the control action itself (if that is the case, the optimal control problem is further complicated).

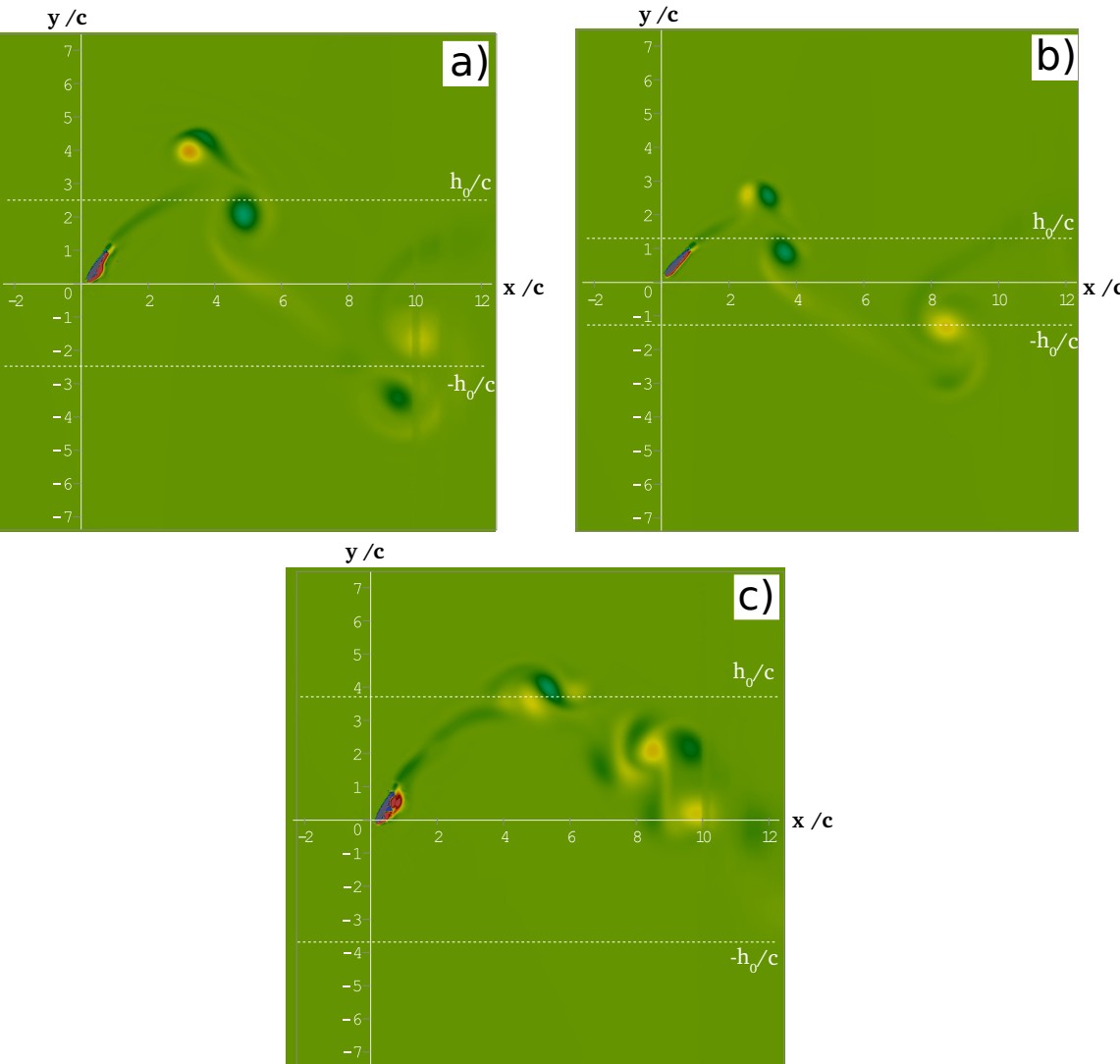

**Figure 6.** Vorticity field, $B = 0.4$ for different times: (**a**) for $h_0/c = 2.5$, $t = 239$ s; (**b**) $h_0/c = 1.25$, $t = 239$ s ; (**c**) $h_0/c = 3.75$, $t = 235$. Red to blue color scale represents positive to negative vorticity.

### 5.2. Effect of B on the Efficiency

The average efficiency $\eta$ and Strouhal number $St$ were tabulated and plotted in order to test the effect of the new dimensionless number $B$ for different values of the (dimensionless) heaving breadth $h_0/c$. Table A4 shows the full set of dimensionless parameters. Figure 7 shows the efficiency curves, in terms of $B$, for different values of $h_0/c$. The markers with dotted lines correspond to high $Re$ with no turbulence model, and the gray markers with no lines are the ones with the turbulence model enabled.

Let us first discuss the dotted curves: as expected, there is a maximum, which is different for each value of $h_0/c$. For the case $h_0/c = 1.25$ (circles), the efficiency increases with B until a maximum is reached near $B = 0.65$. For a larger heaving breadth $h_0/c = 2.5$ (asterisks), power extraction increases, with a maximum efficiency near $\eta = 0.12$, for

$B = 0.38$. In these cases, a larger heaving breadth came along with an increase in dimensionless frequency $St$, as can be inferred from Figure 8, where $St$ is shown as a function of $B$ (same markers). This is the expected behavior, because, with larger breadth, the system can stay more time following the reference angle (very closely, if the control scheme works fine), with a lift force close to maximum, spending relatively small amounts of energy on the control action. Conversely, if the breadth $h_0/c$ is small, then the controller spends more time trying to reach the reference angle (which changes sign more often), with the resulting energy cost and low efficiency.

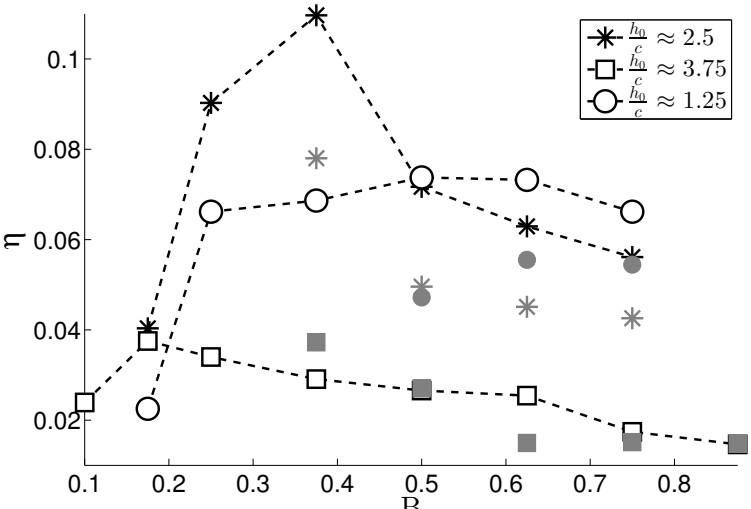

**Figure 7.** $Re = 2 \times 10^4$. $\eta$ vs. $B$. Markers correspond to different values of $h_0/c$: circles: $h_0/c = 1.25$; squares: $h_0/c = 3.75$; asterisks: $h_0/c = 2.5$. Gray markers correspond to same cases with turbulence model applied.

However, if we further increase to $h_0/c = 3.75$, $\eta$ now decreases, as well as the Strouhal number $St$. Careful inspection of the simulation results in time and space makes clear that the system does not recover from dynamic stall (see Figure A1), so the heaving speed is low. This causes the power to decrease, as well as the Strouhal number, as shown in Figure 8.

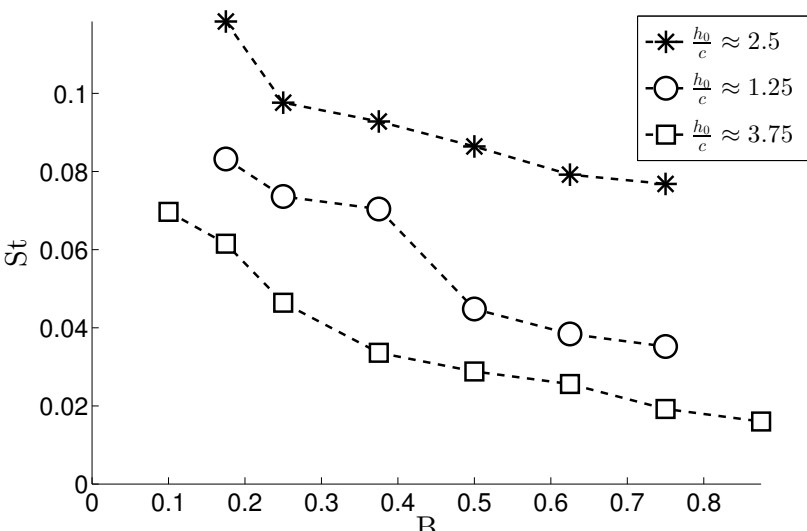

**Figure 8.** $Re = 2 \times 10^4$: $St$ vs. $B$ for different values of $\frac{h_0}{c}$.

For the turbulent cases (gray markers), the efficiency is smaller (as expected), following the same trend as the cases with no turbulent model (dotted lines), with the exception of $h_0/c = 3.75$ (squares). There, the effect of turbulence does not seem to decrease the efficiency (for $B > 0.5$), which is, in both cases, very small due to dynamic stall, as discussed in the previous paragraph. This last curve lies close to the results with no turbulent model.

The set of dimensionless numbers of the foil presented here corresponds to a prototype that fits the laboratory facilities. This prototype is currently being built in order to validate the numerical results with experiments (in a currents channel).

As already mentioned, this controller is not an optimal one, and it may be substantially improved. For instance, the largest contribution in the control energy action amounts to switching the foil direction once $h_0$ is reached. If this was done passively, then the efficiency would reach values above 16%. That can be estimated simply by removing the control effort at the turning region around $h_0$, supposing that a passive mechanism turns the foil back when some $h_0$ is reached. Figure 9 shows such estimation, in the same format as in Figure 7. Here, the maximum efficiency can increase from $\eta = 11\%$ to $\eta = 16\%$. This small exercise makes clear that, for $h_0/c = 1.25$, the energy expense for the abrupt "turn around" of the foil is quite significant.

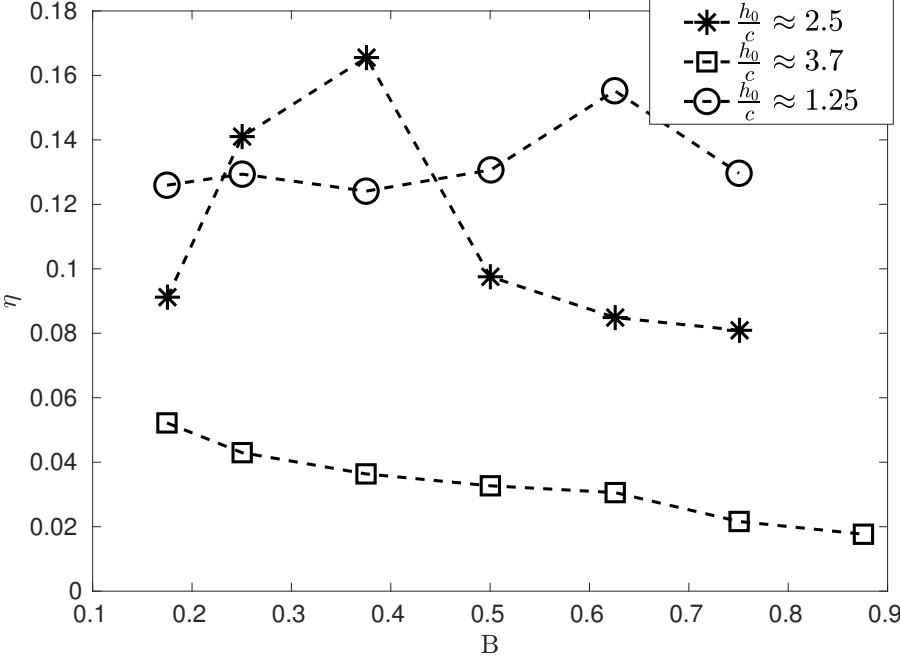

**Figure 9.** $\eta$ vs. $B$. Same as in Figure 7, if the "turn around" of the foil was made passively ($Re = 2 \times 10^4$). The markers correspond to different values of $h_0/c$: circles: $h_0/c = 1.25$; squares: $h_0/c = 3.75$; and, asterisks: $h_0/c = 2.5$.

For the low Reynolds number case ($Re = 1000$), Figure 10 shows the results in the same format as Figure 7 (same markers). The efficiency is much lower, but the behavior of the system is very consistent, and the maxima located at the same $B$ for each $h_0/c$. In this case, for the $h_0/c = 3.75$, the efficiency does not drop so drastically, and the curve is rather flat, as was the case for large $Re$. Note that the power generation is so low that the control effort can overcome it, which results in very low efficiencies (even a case with loss of energy).

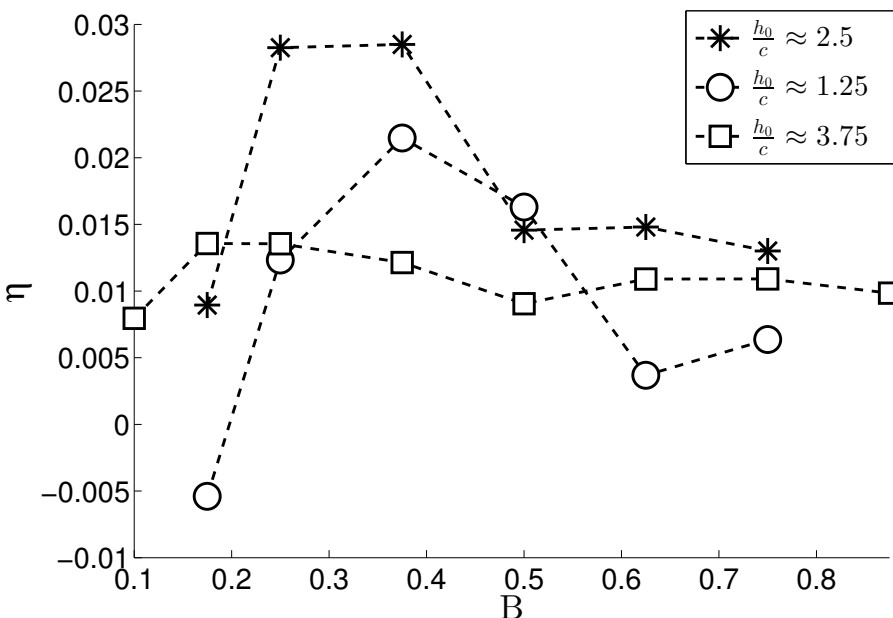

**Figure 10.** $Re = 1000$. $\eta$ vs. $B$. Markers correspond to different values of $h_0/c$: circles: $h_0/c = 1.25$; squares: $h_0/c = 3.75$; asterisks: $h_0/c = 2.5$.

For completeness, Figures 8 and 11 show the $St$ in terms of $h_0/c$ for a large (no turbulence) and low Reynolds number, respectively, same markers. The Strouhal number behaves in a very similar way in both cases. $St$ decreases monotonically with $B$, and the curves do not cross.

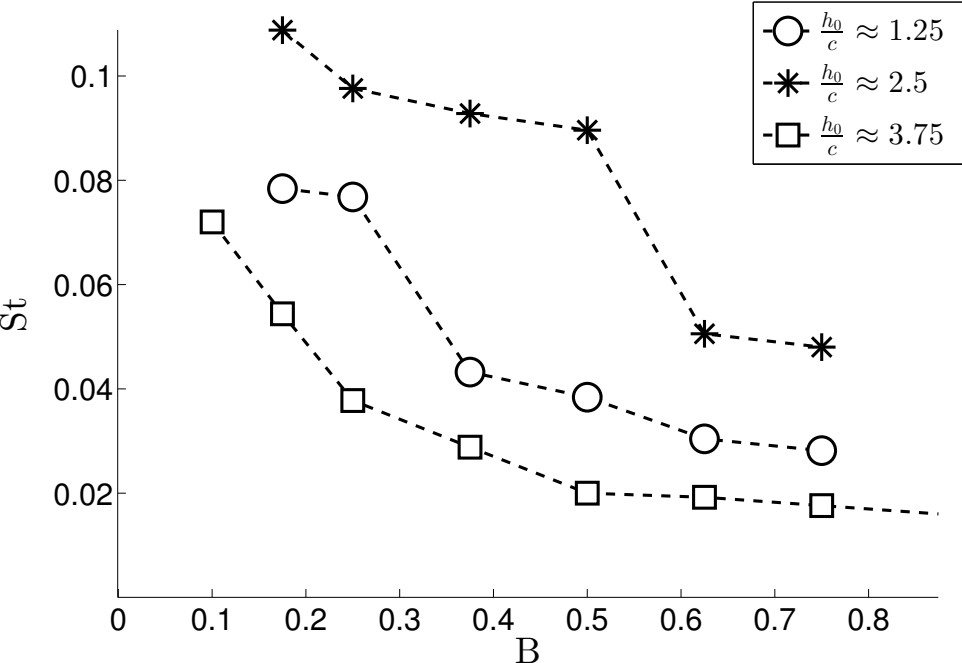

**Figure 11.** $Re = 1000$: $St$ vs. $B$ for different values of $\frac{h_0}{c}$.

For the case where the maximum efficiency was found ($B = 0.375$), a comparison can be made between the three different values of $h_0/c$, in terms of unused energy, control energy, and net harnessed energy. Table A5 shows this. This comparison was possible, because the energy was integrated through a large number of cycles. The total integration time was chosen, such as to make the average energy independent of the total time. These results may vary according to the control scheme used in the closed-loop control, as already

mentioned in previous sections. It is clear from this representation that the best compromise (among the very limited set of dimensionless values we tested) between the harvested energy and that spent in the control action corresponds to $h_0/c = 2.5$.

## 6. Conclusions and Perspectives

A fully coupled numerical simulation that comprises the fluid motion solver, a mechanical system driven by an (immersed) oscillating foil and a feedback control loop was implemented while using a modified version of the open source software OpenFOAM. A proportional feedack control law that follows a reference pitch angle of attack was tested in order to attain maximum lift (based on the static airfoil characteristic curves). A dimensionless parameter, named *B*, which can be interpreted as a comparison between the stiffness of the generator and the control effort, can be used to find the conditions for a maximum efficiency. However, given the absence of analytical solutions of the optimal control problem, more elaborated control strategies and more comprehensive parametric studies are still necessary.

The numerical experiments showed that the phenomenon of dynamic stall may become very important in terms of energy extraction, and they should be taken into account for the design of closed-loop pitch control schemes. Moreover, if the control overshoots, it is likely to trigger dynamic stall.

Among the lessons learned lies the fact that the control effort may become expensive in terms of efficiency, and practical implementations may include mechanisms to switch the heaving direction of the foil passively (in this particular case, increasing the efficiency up to 16%), or, instead of pitching the entire airfoil, one could implement an aileron.

Another important limitation that was evidenced by these numerical experiments is that, whenever the heaving velocity becomes important when compared with the fluid velocity $U$, the lift force (and power) starts to decrease due to the term $\arctan(\dot{y}/U)$ shown in Equation (9).

The proposed scheme allows for the implementation of more elaborated control strategies, like multiple-input-multiple-output (MIMO), optimal, or adaptive control. For instance, it would be possible to use such simulations to infer an equivalent reduced system in order to replace the actual fluid-solid-body interaction. This would open the possibility of synthesis of an optimal controller (maximize efficiency), use predictive control strategies [55], or even the use of neural networks and artificial intelligence algorithms for control purposes, as suggested in [56] for a wide range of applications.

We are currently testing the feedback loop in three-dimensions (3D) while using Lattice–Boltzman methods, and we have found very similar effects of the feedback law on the system's performance. In the present study, two dimensional laminar flow simulations were performed in order to assess the effect of the feedback loop on the efficiency of the oscillating device. It is clear that under operating conditions turbulence, finite wingspan, ground, and three dimensional effects will have a significant impact on efficiency. We are currently working to address such effects as well as to include experimental observations in a water tunnel; our results will be published in the near future.

**Author Contributions:** Conceptualization, B.F.-E., D.B.-T. and C.M.; writing—original draft preparation, D.B.-T.; writing—review and editing, B.F.-E. and C.M. All authors have read and agreed to the published version of the manuscript.

**Funding:** This research was funded by UNAM DGAPA PAPIIT grant No. IA102016.

**Institutional Review Board Statement:** Not applicable.

**Informed Consent Statement:** Not applicable.

**Data Availability Statement:** The data presented in this study are available on request from the corresponding author.

**Acknowledgments:** The authors respectfully acknowledge the support of CONACYT grant FORDECYT-PRONACES 21088 and UNAM DGAPA PAPIIT grant No. IA102016. D.B. thanks CONACYT for the support during his Ph. D. studies. C. Málaga acknowledges the support from UNAM-PASPA during his sabbatical stay at LIPC-UNAM.

**Conflicts of Interest:** The authors declare no conflict of interest. The funders had no role in the design of the study, however they strongly encourage and support the publication of the results.

## Appendix A. Tables and Figures

**Table A1.** Numerical studies.

| Authors | Year | AR | Type | Foil | *Re* | $\eta_{max}$ |
|---------|------|-----|------|------|------|--------------|
| Kinsey and Dumas | 2012 | 2D | Prescribed & tandem | NACA0015 | $5.0 \times 10^5$ | 0.63 |
| Platzer et al. | 2010 | 2D | Fully Passive prescribed y tandem | NACA0014 | $2.0 \times 10^4$ | 0.54 |
| Ashraf et al. | 2011 | 2D | Prescribed & tandem | NACA0014 | $2.0 \times 10^4$ | 0.54 |
| Young et al. | 2013 | 2D | Fully passive | NACA0012 | $1.1 \times 10^3$–$1.1 \times 10^6$ | 0.41 |
| Campobasso et al. | 2013 | 2D | Prescribed | NACA0015 | $1.1 \times 10^3$–$1.5 \times 10^6$ | 0.40 |
| Le et al. | 2013 | 2D | Prescribed | Biomimetic | $9 \times 10^4$ | 0.39 |
| Ashraf et al. | 2009 | 2D | Prescribed | NACA0012 | 1100 | 0.38 |
| Shimizu et al. | 2008 | 2D | Semi-passive open-loop | NACA0012 | $4.62 \times 10^5$ | 0.35 |
| This study | 2019 | | Semi-passive closed-loop | NACA0015 | $2.0 \times 10^5$ | 0.12 |

**Table A2.** Experimental studies.

| Authors | Year | Type | Foil | *Re* | $\eta_{max}$ |
|---------|------|------|------|------|--------------|
| Kinsey et al. | 2011 | Fully passive | NACA0015 | $5 \times 10^5$ | 0.4 |
| Kinsey and Dumas | 2010 | Fully passive | NACA0015 | $5 \times 10^5$ | 0.4 |
| Simpson et al. | 2009 | Prescribed | NACA0012 | $1.38 \times 10^4$ | 0.32 |
| Huxham et al | 2012 | Semi-passive | NACA0015 | $4.5 \times 10^4$ | 0.24 |
| Lindsey, Jones et al. | 2003 | Fully passive | NACA0014 | $2.2 \times 10^4$ | 0.23 |
| McKinney and DeLaurier | 1981 | Fully passive | NACA0012 | $8.5 \times 10^4$–$1.1 \times 10^5$ | 0.17 |

**Table A3.** Mesh parameters. The initial-final-refinement level indicates how refinement evolves on the different mesh parts. Larger values indicate more refinement. The first and second digits indicate the two levels of refinement closest to the foil surface. the last digit indicates how many levels of refinement exist in the refinement box (large square containing the foil in Figure 2. The test parameter was energy output averaged through a large number of cycles (more than 25).

| Name | Number of Cells | Initial-Final-Refinement Level | Average Energy (J) |
|------|-----------------|-------------------------------|--------------------|
| M1 | 18,040 | 3-4-1 | 5.65 |
| M2 | 47,440 | 3-4-3 | 5.649 |
| M3 | 290,326 | 4-4-1 | 5.654 |
| M4 | 291,286 | 4-4-3 | 5.654 |
| M5 | 322,606 | 4-5-3 | 5.655 |

**Table A4.** Dimensionless parameters matrix.

| ID | $\eta$ [%] | $Re$ | $\frac{h_0}{c}$ | $\Pi_1$ | $\Pi_2$ | $\Pi_3$ | $\Pi_4$ | $St$ | $B$ |
|---|---|---|---|---|---|---|---|---|---|
| 1 | 4.03 | $2 \times 10^4$ | 2.5 | 0.0937 | $5.12 \times 10^{-6}$ | 0.0437 | 0.25 | $8.32 \times 10^{-2}$ | 0.175 |
| 2 | 9.03 | $2 \times 10^4$ | 2.5 | 0.0937 | $5.12 \times 10^{-6}$ | 0.0625 | 0.25 | $7.36 \times 10^{-2}$ | 0.25 |
| 3 | 10.97 | $2 \times 10^4$ | 2.5 | 0.0937 | $5.12 \times 10^{-6}$ | 0.0937 | 0.25 | $7.04 \times 10^{-2}$ | 0.375 |
| 4 | 7.17 | $2 \times 10^4$ | 2.5 | 0.0937 | $5.12 \times 10^{-6}$ | 0.1250 | 0.25 | $4.48 \times 10^{-2}$ | 0.5 |
| 5 | 6.29 | $2 \times 10^4$ | 2.5 | 0.0937 | $5.12 \times 10^{-6}$ | 0.1562 | 0.25 | $3.84 \times 10^{-2}$ | 0.625 |
| 6 | 5.61 | $2 \times 10^4$ | 2.5 | 0.0937 | $5.12 \times 10^{-6}$ | 0.1875 | 0.25 | $3.52 \times 10^{-2}$ | 0.75 |
| 7 | 2.25 | $2 \times 10^4$ | 1.25 | 0.0937 | $5.12 \times 10^{-6}$ | 0.0437 | 0.25 | $11.84 \times 10^{-2}$ | 0.175 |
| 8 | 6.62 | $2 \times 10^4$ | 1.25 | 0.0937 | $5.12 \times 10^{-6}$ | 0.0625 | 0.25 | $9.76 \times 10^{-2}$ | 0.25 |
| 9 | 6.86 | $2 \times 10^4$ | 1.25 | 0.0937 | $5.12 \times 10^{-6}$ | 0.0937 | 0.25 | $9.28 \times 10^{-2}$ | 0.375 |
| 10 | 7.38 | $2 \times 10^4$ | 1.25 | 0.0937 | $5.12 \times 10^{-6}$ | 0.1250 | 0.25 | $8.64 \times 10^{-2}$ | 0.5 |
| 11 | 7.32 | $2 \times 10^4$ | 1.25 | 0.0937 | $5.12 \times 10^{-6}$ | 0.1562 | 0.25 | $7.92 \times 10^{-2}$ | 0.625 |
| 12 | 6.62 | $2 \times 10^4$ | 1.25 | 0.0937 | $5.12 \times 10^{-6}$ | 0.1875 | 0.25 | $7.68 \times 10^{-2}$ | 0.75 |
| 13 | 2.39 | $2 \times 10^4$ | 3.75 | 0.0937 | $5.12 \times 10^{-6}$ | 0.0250 | 0.25 | $6.97 \times 10^{-2}$ | 0.1 |
| 14 | 3.75 | $2 \times 10^4$ | 3.75 | 0.0937 | $5.12 \times 10^{-6}$ | 0.0437 | 0.25 | $6.15 \times 10^{-2}$ | 0.175 |
| 15 | 3.40 | $2 \times 10^4$ | 3.75 | 0.0937 | $5.12 \times 10^{-6}$ | 0.0625 | 0.25 | $4.64 \times 10^{-2}$ | 0.25 |
| 16 | 2.91 | $2 \times 10^4$ | 3.75 | 0.0937 | $5.12 \times 10^{-6}$ | 0.0937 | 0.25 | $3.36 \times 10^{-2}$ | 0.375 |
| 17 | 2.66 | $2 \times 10^4$ | 3.75 | 0.0937 | $5.12 \times 10^{-6}$ | 0.1250 | 0.25 | $2.88 \times 10^{-2}$ | 0.5 |
| 18 | 2.54 | $2 \times 10^4$ | 3.75 | 0.0937 | $5.12 \times 10^{-6}$ | 0.1562 | 0.25 | $2.56 \times 10^{-2}$ | 0.625 |
| 19 | 1.74 | $2 \times 10^4$ | 3.75 | 0.0937 | $5.12 \times 10^{-6}$ | 0.1875 | 0.25 | $1.92 \times 10^{-2}$ | 0.75 |
| 20 | 1.46 | $2 \times 10^4$ | 3.75 | 0.0937 | $5.12 \times 10^{-6}$ | 0.2187 | 0.25 | $1.60 \times 10^{-2}$ | 0.875 |
| 1 | 0.009 | $1 \times 10^3$ | 2.5 | 0.0937 | $9.8 \times 10^{-3}$ | 0.0437 | 0.25 | $7.84 \times 10^{-2}$ | 0.175 |
| 2 | 0.028 | $1 \times 10^3$ | 2.5 | 0.0937 | $9.8 \times 10^{-3}$ | 0.0625 | 0.25 | $8.0 \times 10^{-2}$ | 0.25 |
| 3 | 0.016 | $1 \times 10^3$ | 2.5 | 0.0937 | $9.8 \times 10^{-3}$ | 0.0937 | 0.25 | $5.2 \times 10^{-2}$ | 0.375 |
| 4 | 0.015 | $1 \times 10^3$ | 2.5 | 0.0937 | $9.8 \times 10^{-3}$ | 0.1250 | 0.25 | $3.6 \times 10^{-2}$ | 0.5 |
| 5 | 0.028 | $1 \times 10^3$ | 2.5 | 0.0937 | $9.8 \times 10^{-3}$ | 0.1562 | 0.25 | $2.56 \times 10^{-2}$ | 0.625 |
| 6 | 0.024 | $1 \times 10^3$ | 2.5 | 0.0937 | $9.8 \times 10^{-3}$ | 0.1875 | 0.25 | $2.4 \times 10^{-2}$ | 0.75 |
| 7 | -0.005 | $1 \times 10^3$ | 1.25 | 0.0937 | $9.8 \times 10^{-3}$ | 0.0437 | 0.25 | $10.66 \times 10^{-2}$ | 0.175 |
| 8 | 0.012 | $1 \times 10^3$ | 1.25 | 0.0937 | $9.8 \times 10^{-3}$ | 0.0625 | 0.25 | $9.44 \times 10^{-2}$ | 0.25 |
| 9 | 0.021 | $1 \times 10^3$ | 1.25 | 0.0937 | $9.8 \times 10^{-3}$ | 0.0937 | 0.25 | $8.96 \times 10^{-2}$ | 0.375 |
| 10 | 0.016 | $1 \times 10^3$ | 1.25 | 0.0937 | $9.8 \times 10^{-3}$ | 0.1250 | 0.25 | $11.2 \times 10^{-2}$ | 0.5 |
| 11 | 0.004 | $1 \times 10^3$ | 1.25 | 0.0937 | $9.8 \times 10^{-3}$ | 0.1562 | 0.25 | $4.8 \times 10^{-2}$ | 0.625 |
| 12 | 0.006 | $1 \times 10^3$ | 1.25 | 0.0937 | $9.8 \times 10^{-3}$ | 0.1875 | 0.25 | $4.64 \times 10^{-2}$ | 0.75 |
| 13 | 0.008 | $1 \times 10^3$ | 3.75 | 0.0937 | $9.8 \times 10^{-3}$ | 0.0250 | 0.25 | $6.88 \times 10^{-2}$ | 0.1 |
| 14 | 0.013 | $1 \times 10^3$ | 3.75 | 0.0937 | $9.8 \times 10^{-3}$ | 0.0437 | 0.25 | $5.44 \times 10^{-2}$ | 0.175 |
| 15 | 0.014 | $1 \times 10^3$ | 3.75 | 0.0937 | $9.8 \times 10^{-3}$ | 0.0625 | 0.25 | $3.52 \times 10^{-2}$ | 0.25 |
| 16 | 0.012 | $1 \times 10^3$ | 3.75 | 0.0937 | $9.8 \times 10^{-3}$ | 0.0937 | 0.25 | $3.2 \times 10^{-2}$ | 0.375 |
| 17 | 0.009 | $1 \times 10^3$ | 3.75 | 0.0937 | $9.8 \times 10^{-3}$ | 0.1250 | 0.25 | $1.76 \times 10^{-2}$ | 0.5 |
| 18 | 0.011 | $1 \times 10^3$ | 3.75 | 0.0937 | $9.8 \times 10^{-3}$ | 0.1562 | 0.25 | $1.92 \times 10^{-2}$ | 0.625 |
| 19 | 0.011 | $1 \times 10^3$ | 3.75 | 0.0937 | $9.8 \times 10^{-3}$ | 0.1875 | 0.25 | $1.6 \times 10^{-2}$ | 0.75 |
| 20 | 0.010 | $1 \times 10^3$ | 3.75 | 0.0937 | $9.8 \times 10^{-3}$ | 0.2187 | 0.25 | $1.44 \times 10^{-2}$ | 0.875 |

**Table A5.** Control energy, unused energy and net harnessed energy for different $\frac{h_0}{c}$.

| $\frac{h_0}{c}$ | Unused Energy | Net Harnessed Energy | Control Energy Consumption |
|---|---|---|---|
| 2.5 | 83.8% | 10.6% | 5.7% |
| 1.25 | 87.3% | 6.9% | 5.8% |
| 3.75 | 95.9% | 3.2% | 0.9% |

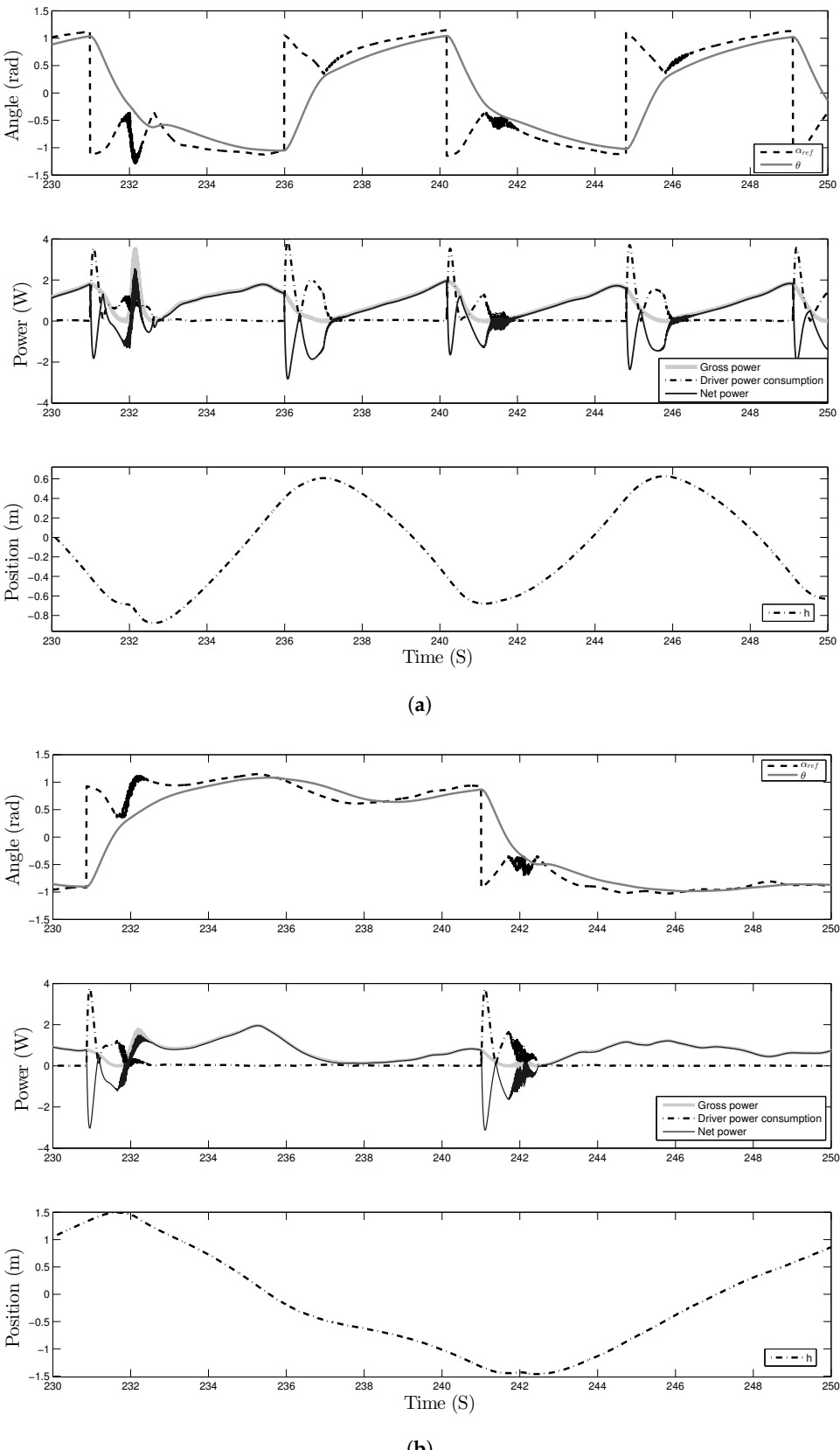

**Figure A1.** System response for different $h_0/c$. (**a**) $h_0/c = 1.25$, (**b**) $h_0/c = 3.25$, as in Figure 5.

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
