# Peer review of "Numerical Study of an Oscillating-Wing Wingmill for Ocean Current Energy Harvesting: Fluid-Solid-Body Interaction with Feedback Control"

_jmse, doi:10.3390/jmse9010023_

Round 1
Reviewer 1 Report
This is an interesting paper regarding an oscillating foil in water for energy production. Presentation is good, there are some spelling errors, but overall language is fine. However, I have some critical remarks regarding the numerical model. More time should be spend to refine that, in particular turbulence and its modelling (in particular stall and transition) is not captured well enough with your high-Reynolds RANS approach.
- Line 40: unnecessary bracket
- Table A4 and A5: correct formatting of Reynolds-number values for Young and Campobasso (A4) and all authors in A5
- Line 67: no comma before ‘that’
- Line 68: ‘center’ instead of ‘centre’ as in the rest of the document
- Line 112: Please also provide the angle in degree
- Line 127: ‘… numerical method … called PIMPLE’ – PIMPLE is only the pressure-velocity coupling algorithm for solving incompressible flows. In general more information regarding the numerical model is needed, e.g. which discretization schemes are used, which solver is chosen, time step size and Courant number, …
- Line 159: Please add which tool you used for creating the mesh
- Line 169: ‘total Power’ – no capital ‘P’
- Line 185: Would be better to have also investigated meshes with higher cell counts. All but one mesh have basically the same cell count. It is better to keep the relative refinement ratio the same and slowly increase the resolution of the background mesh
- Line 207 - 2014: Several words with capital letters for no reason, e.g. ‘Plant’, ‘Data Extraction’, ‘The Reference Angle’
- Figure 4: Some words with capital letter, some with small letters
- Line 228: no column after ‚inertia‘
- Line 265: ‘A RANS-based turbulence model (k-omega-SST) was chosen, which can be used as a Low-Re turbulence model’ – your y+ value is 50, so you are basically using a high-Reynolds number model and the k-omega part in the turbulence model has basically no influence. You should describe that more accurately.
- Line 269: Have you tested different y+ values for your mesh? In airfoil simulations y+ has a tremendous effect on lift coefficient, in particular close to stall the critical angle is barely correct for such high y+ values as in your case.
- Line 280 – 301: many words with capital letters, e.g. ‘Gross Power’, ‘Generator Power’, ‘Power Consumption’. Please correct that
- Tables in appendix are in reverse order
- Figure 7: The influence of turbulence modelling (or no turbulence) has a significant effect on the results. More time should be spend on the numerical model to account for this, e.g. higher near-wall mesh resolution in combination with a transition turbulence model such as the Langtry-Menter k-omega SST model
Reviewer 2 Report
- The usage of the word “wingmill” in the paper title is questionable. In the introduction section, it is said that the idea of harvesting energy with oscillating foils (called wingmills) was first presented in 1981, [5]. However, the title of [5] is “an oscillating-wing windmill”, not “wingmill”.
-
Eq(9) is not clear to me. The term sign(\theta_ref) may be important but it does not agree with Eq (4), (7), and (8).
- Figure 4 is not accurate. For example, the term \dot{h(t)} is not correct? Also the +- sign is reversed?
- The major problem is: we expect e(t) goes to zero (from control perspective), which implies \theta(t)=\alpha_ref and that \alpha(t)=\theta_ref. This is very troublesome to me.